# A Receptor Integrin β1 Promotes Infection of Avian Metapneumovirus Subgroup C by Recognizing a Viral Fusion Protein RSD Motif

**DOI:** 10.3390/ijms25020829

**Published:** 2024-01-09

**Authors:** Yongqiu Cui, Siting Li, Weiyin Xu, Yeqiu Li, Jiali Xie, Dedong Wang, Jinshuo Guo, Jianwei Zhou, Xufei Feng, Lei Hou, Jue Liu

**Affiliations:** 1College of Veterinary Medicine, Yangzhou University, Yangzhou 225009, China; cuiyongqiu97@163.com (Y.C.); sitingli999@163.com (S.L.); xuxuwy@163.com (W.X.); liyeqiu1996@163.com (Y.L.); xjl999736@163.com (J.X.); wddyzu@163.com (D.W.); jane9407@163.com (J.G.); jwzhou@yzu.edu.cn (J.Z.); xffeng@yzu.edu.cn (X.F.); 2Jiangsu Co-Innovation Center for Prevention and Control of Important Animal Infectious Diseases and Zoonoses, Yangzhou University, Yangzhou 225009, China

**Keywords:** avian metapneumovirus subgroup C, fusion protein, integrin β1, viral entry, avian and human cells

## Abstract

Avian metapneumovirus subgroup C (aMPV/C) causes respiratory diseases and egg dropping in chickens and turkeys, resulting in severe economic losses to the poultry industry worldwide. Integrin β1 (ITGB1), a transmembrane cell adhesion molecule, is present in various cells and mediates numerous viral infections. Herein, we demonstrate that ITGB1 is essential for aMPV/C infection in cultured DF-1 cells, as evidenced by the inhibition of viral binding by EDTA blockade, Arg-Ser-Asp (RSD) peptide, monoclonal antibody against ITGB1, and ITGB1 short interfering (si) RNA knockdown in cultured DF-1 cells. Simulation of the binding process between the aMPV/C fusion (F) protein and avian-derived ITGB1 using molecular dynamics showed that ITGB1 may be a host factor benefiting aMPV/C attachment or internalization. The transient expression of avian ITGB1-rendered porcine and feline non-permissive cells (DQ cells and CRFK cells, respectively) is susceptible to aMPV/C infection. Kinetic replication of aMPV/C in siRNA-knockdown cells revealed that ITGB1 plays an important role in aMPV/C infection at the early stage (attachment and internalization). aMPV/C was also able to efficiently infect human non-small cell lung cancer (A549) cells. This may be a consequence of the similar structures of both metapneumovirus F protein-specific motifs (RSD for aMPV/C and RGD for human metapneumovirus) recognized by ITGB1. Overexpression of avian-derived ITGB1 and human-derived ITGB1 in A549 cells enhanced aMPV/C infectivity. Taken together, this study demonstrated that ITGB1 acts as an essential receptor for aMPV/C attachment and internalization into host cells, facilitating aMPV/C infection.

## 1. Introduction

Metapneumovirus (MPV), a negative-sense and single-stranded RNA virus, is classified into two different types, human metapneumovirus (hMPV) and avian metapneumovirus (aMPV) (subfamily *Pneumovirinae*, family *Paramyxoviridae*), which exhibit a propensity for interspecies transmission [1,2,3]. aMPV is a major cause of acute rhinotracheitis in turkeys and is associated with swollen head syndrome in chickens and turkeys [4]; moreover, it can infect other avian species, such as ducks, sparrows, and pigeons [5,6,7,8]. Based on its genetic and antigenic properties, aMPV can be divided into four subgroups: A, B, C, and D [9]. aMPV subgroup C (aMPV/C) was first identified in turkeys in the United States [10,11] and was subsequently detected in other avian species in other countries, such as France [12], Korea [13], China [9,13], and Italy [14]. A recent study found that human metapneumovirus (hMPV) is closely associated with aMPV/C, suggesting that hMPV may have evolved from avian metapneumovirus via zoonotic transmission [15]. In addition, aMPV/C has been shown to infect mice, where it can replicate in the lungs and cause pathological changes, including an increased number of inflammatory cells [16]. These findings may implicate aMPV/C as a potential zoonotic pathogen, possessing public health significance.

aMPV/C genome encodes eight proteins: nucleocapsid protein (N), phosphoprotein (P), matrix protein (M), fusion protein (F), second matrix protein (M2), small hydrophobic protein (SH), attachment glycoprotein (G), and RNA-dependent RNA polymerase (L) [4]. The virus entering the cell represents the beginning of infection. Paramyxovirus entry into host cells involves the fusion of a viral envelope with the cell membrane [17]. Most viruses only utilize one protein for cellular entry and fusion such as HA for the influenza virus, spike protein for coronavirus, and glycoprotein (G) for VSV [18,19,20]. However, paramyxoviruses, including aMPV, human metapneumovirus (hMPV), and human respiratory syncytial virus, can use three proteins (F, G, and SH) to help its cellular entry and fusion, the G protein mainly plays the role of mediating the adsorption of metapneumovirus to cells, and SH contributes to cell–cell fusion; meanwhile, the F protein can bind to receptors that mediate the fusion of the viral envelope with the cell membrane without relying on G and SH proteins [21,22,23,24]. The F protein is synthesized from the precursor protein F0, which is processed into two disulfide-linked F1 and F2 subunits immediately after activation [21]. It is generally acknowledged that the binding of fusion proteins to the cell surface receptor(s) mediates conformational changes in the F protein, which, subsequently, triggers membrane fusion [25,26,27,28].

Integrins, the major receptors, are involved in the adhesion of cells to extracellular matrix proteins and contribute crucially to vertebrate cell–cell adhesion [29]. In addition, integrins induce the connections between the transmembrane and the cytoskeleton to activate intracellular signals. Integrins are receptors for various viruses and bacteria [29,30]. For example, integrins modulate aMPV/B and hMPV by facilitating protein-induced cell–cell fusion and viral infection, in which F proteins of aMPV/B and hMPV can induce membrane fusion, requiring the binding of Arg-Asp-Asp (RDD) or Arg-Gly-Asp (RGD) motifs to integrin [31,32]. Furthermore, the Arg-Ser-Asp (RSD) motif mediates membrane fusion by binding to integrins in the case of viral entry [33]. The high expression of integrins on the cell membrane surface, high conservation of integrins among different species, and presence of an RSD motif in aMPV/C indicate that integrins might be potential receptors for aMPV/C infection.

In the present study, we verified the hypothesis that integrins act as functional receptors for aMPV/C. The divalent cation chelator ethylenediaminetetraacetic acid (EDTA), which decreases integrin–ligand binding, diminishes aMPV/C infectivity. Functional blocking of integrin β1 (ITGB1)-specific antibodies inhibit aMPV/C infection. The transfection of avian-derived ITGB1 cDNA into poorly permissive cells conferred aMPV/C infectivity, and aMPV/C used ITGB1 to enhance its ability to infect human lung cells. The downregulation of ITGB1 expression by siRNA diminished aMPV/C infection. Collectively, these data provide strong evidence that aMPV/C can use ITGB1 as a cellular receptor to mediate viral entry into avian cells and that aMPV/C can also infect human lung cells via binding to human-derived ITGB1.

## 2. Results

### 2.1. aMPV/C F Protein Contains a Highly Conserved RSD Motif

We compared the sequences of the full-length F gene in all aMPV/C isolates collected from the NCBI GenBank database (https://www.ncbi.nlm.nih.gov/, accessed on 31 March 2023). An Arg-Ser-Asp (RSD) motif at residues 329–331 was strictly conserved in all aMPV/C F proteins (Appendix A). To determine whether RSD motif exposure was lateral to the F protein, its structure was constructed using AlphaFold2 (Appendix A). The resolved F protein structure of aMPV/C showed that the F protein comprised 527 amino acids, with a molecular formula of C2568H4150N690O783S26, and contained 8217 atoms. The relative molecular mass was 58,052.91, the theoretical isoelectric point was 6.84, and the instability coefficient of the F protein was 36.34. The aliphatic coefficient was 97.09, and the average hydrophilicity was 0.060. The secondary structure of the F protein mainly consisted of 205 α helices (h, 38.18%), 129 extensions (e, 24.02%), 60 β turns (t, 11.17%), and 143 random helices (c, 26.63%). Meanwhile, a library comprising more than 30,000 protein receptors (www.bindingdb.org, accessed on 31 March 2023) was used to simulate molecular docking with the aMPV/C F protein RSD motif. The results indicated that the RSD can bind to integrins, leading to the hypothesis that integrins may function as receptors for aMPV/C.

### 2.2. EDTA and Integrin β1-Specific Antibodies Reduce aMPV/C Infectivity

Divalent cations, such as Ca^2+^ and Mg^2+^ [30], are required for integrin–ligand interactions, and previous studies have indicated that EDTA is a divalent cation chelator that can inhibit such interactions [32]. Firstly, the cell viability in EDTA-treated cells was analyzed using an MTT assay. The results showed that the EDTA treatment did not affect the cell viability up to 2.5 mM (Figure 1A). Subsequently, to verify the effect of EDTA on aMPV/C infection, the DF-1 cells treated with EDTA were infected with aMPV/C, and aMPV/C replication was analyzed using Western blotting, viral titers, and viral genome copy numbers. The results showed that EDTA inhibited aMPV/C replication in a dose-dependent manner, which was characterized by a reduction in N protein expression, viral titers, and viral genome copy numbers (Figure 1B–D). Moreover, further validation was performed using an indirect immunofluorescence assay. The results showed that the number of positive cells with fluorescence decreased with increasing EDTA concentration (Figure 1E). Thus, we confirmed that EDTA exerts an inhibitory effect on aMPV/C replication, supporting the positive regulatory role of integrins in aMPV/C infection.

To identify whether the integrins were required for aMPV/C infection, DF-1 cells were incubated with anti-integrin antibodies to block the binding of RSD and integrins, followed by aMPV/C infection, and the F protein expression, virus replication, and viral titer were analyzed. The results showed that specific antibodies against αv and β1 integrins, separately, caused the significant blockage for aMPV/C replication compared to the other antibodies (anti-α6, α2, and β3 integrins antibodies), exhibited approximately 40% blockage rates for aMPV/C replication, and the combination of anti-αv with anti-β1 integrin antibodies resulted in at least 60% blockage rates, whereas specific antibodies against α6, α2, and β3 integrins showed a minimal inhibitory effect, suggesting that anti-αv and/or anti-β1 integrin antibodies caused the significant blockage for aMPV/C replication compared to the other antibodies (anti-α6, α2, and β3 integrins antibodies) (Figure 2). Thus, these data indicated that αvβ1 integrin was involved in the infection of aMPV/C.

### 2.3. RSD-Specific Peptide Reduces aMPV/C Infection

The previous study found that the RSD motif in the F protein was a key motif for binding integrin [34,35]. To further evaluate the role of the RSD motif in virus replication, the RSD or Arg-Ser-Glu (RGE) peptides were synthesized to analyze the blocking effect. Firstly, the cell viability of the RSD or Arg-Ser-Glu (RGE) peptides was analyzed using MTT assays. The results did not show significant cytotoxicity in the cells treated with the RSD or Arg-Ser-Glu (RGE) peptides (Figure 3A). Subsequently, the DF-1 cells were incubated with the GRSDSP (integrin-binding peptide) or GRGESP (the control peptide), followed by aMPV/C infection. The results showed that GRSDSP treatment inhibited aMPV/C infection in a dose-dependent manner, whereas the GRGESP peptide had no significant inhibitory effect (Figure 3B–E), suggesting that the RSD motif was required for aMPV/C infection. The inhibitory effect of the RSD peptide was modest, but the blockade magnitude of the infectivity was similar to that of other viruses engaging integrins [33].

### 2.4. aMPV/C Can Use Avian ITGB1 as a Receptor for Cell Entry

To investigate the role of the ITGB1 interaction with aMPV/C, molecular docking methods were used to simulate the binding between chicken ITGB1 (cITGB1) and the aMPV/C F protein. As shown in Figure 4, the binding energy of the RSD motif to cITGB1 was −5.5 kcal/mol, and the formation of hydrogen bonds and hydrophobic forces benefited from the interaction between the RSD motif and ITGB1, indicating that the F protein was able to bind to cITGB1. During this interaction process, Arg (R) formed hydrogen bonds with Leu693, Asp691, and Gln689, with lengths of 3.53 Å, 3.59 Å, 3.44 Å, and 3.71 Å, respectively; Ser (S) formed hydrogen bonds with Leu693 and Ser694 with lengths of 3.87 Å and 2.90 Å, respectively; and Asp (D) formed hydrogen bonds with His695, Cys645, Val644, and Gly643, with lengths of 3.34 Å, 3.13 Å, 2.95 Å, and 3.40 Å, respectively (Figure 4). The formation of a hydrophobic interaction between the RSD motif and Pro687 increased the ability of the F protein to bind to cITGB1 (Figure 4), suggesting that these residues may be the key active sites for peptides to act on the F protein.

To investigate the effect of integrin expression on aMPV/C infection, we transfected DF-1 cells with cITGB1 plasmids. The overexpression of cITGB1 increased the replication of aMPV/C (Figure 5A–C). The intestinal porcine epithelial (DQ) and feline kidney (CRFK) cell lines were poorly permissive to aMPV/C infection (Figure 5D). Both the DQ cells and CRFK cells transfected with cITGB1 plasmids were infected with aMPV/C, followed by fluorescence analysis. The results showed that the specific fluorescence signals were observed in the poorly permissive cells for aMPV/C replication, suggesting that cITGB1 expression substantially facilitated aMPV/C infection, which indicated that chicken ITGB1 can significantly enhance aMPV infection in DQ and CRFK cells (Figure 5D–F). These results demonstrated that the overexpression of cITGB1 promoted aMPV/C infection.

To determine whether the reduced cITGB1 expression limits aMPV/C infection, DF-1 cells were transfected with siRNAs targeting cITGB1. Transfection with cITGB1 siRNA reduced the cell surface expression of cITGB1 in most cells, whereas nonspecific siRNA transfection did not diminish the expression of cITGB1 (Figure 6A). aMPV/C replication was significantly downregulated compared to the control group (Figure 6B). These results indicated that cITGB1 expression facilitated aMPV/C replication.

Based on ITGB1, as a receptor, located on the surface of cells [29,30], we speculated whether cITGB1 affected aMPV/C attachment or entry and finally regulated aMPV/C replication. To confirm this hypothesis, the treatment of chemical reagent, RSD peptide, or cITGB1 siRNA was used to analyze the effect of cITGB1 on aMPV/C attachment or entry. DF-1 cells treated with EDTA, RSD peptide, or cITGB1 siRNA were infected with aMPV/C at 4 °C for analyzing viral attachment, followed by shifting to 37 °C for analyzing viral entry. The results showed that EDTA, RSD peptide, or cITGB1 siRNA not only significantly decreased aMPV/C attachment but also aMPV/C entry (Figure 7). Therefore, our results demonstrated that cITGB1 was identified as a cell receptor for aMPV/C infection and played a crucial role in the attachment process of aMPV/C.

### 2.5. aMPV/C Can Infect A549 Cells

The genetic relationship between hMPV and aMPV/C and the presumed origin of hMPV in avian species [15] allowed us to investigate whether hMPV-permissive cells, namely, A549, could be infected with aMPV/C. Firstly, the F protein structures between aMPV/C and hMPV were compared. We found that the structures between the RSD motif in aMPV F proteins and the RGD motif in hMPV F proteins were highly similar and were exposed to the surface of F proteins (Appendix A). The previous study found that the RSD motif can also bind to integrin and mediate membrane fusion [33]. To determine whether aMPV/C can infect A549 cells, we employed molecular docking to measure the binding energy between aMPV/C F protein and human ITGB1 (hITGB1). The simulation analysis results indicated that aMPV/C F protein can bind to hITGB1 with a binding energy of −5.2 kcal/mol, accompanied by the formation of hydrogen bonds and hydrophobic forces between RSD peptides and hITGB1 (Appendix A). For example, Arg (R) formed hydrogen bonds with Asp622 and Thr621 with lengths of 3.48 Å and 3.63 Å, respectively; Ser (S) formed hydrogen bonds with Leu637, with lengths of 3.29 Å and 3.03 Å, respectively; and the RSD exhibited hydrophobic interactions with Arg610, Asn710, Asn711, Val639, and Cys636 (Appendix A). Therefore, these results suggested that hITGB1 could recognize the RSD motif. We performed aMPV/C infection experiments in A549 cells to confirm the above results. Remarkably, A549 cells were susceptible to aMPV/C infection without any treatment (Figure 8). Moreover, to determine whether ITGB1 served as a receptor for aMPV/C infection in A549 cells, we transfected human and avian ITGB1 plasmids into A549 cells. The results showed that the expression of cITGB1 and hITGB1 in A549 cells substantially enhanced the efficiency of aMPV/C infection (Figure 8). These results indicate that aMPV/C can also use human ITGB1 molecules as a receptor to infect cells of human origin.

## 3. Discussion

The virus must rely on the assistance of receptors to enter cells; however, the receptor for aMPV/C is currently unclear. In this study, simulation of the aMPV/C F protein using an artificial intelligence algorithm (AlphaFold2) revealed the presence of an RSD motif on the outside of the F protein, indicating that the RSD motif may facilitate receptor binding. Peptide-blocking experiments showed that the RSD motif reduced the aMPV/C infection. Molecular dynamics was also used to simulate the combination of the aMPV/C F protein and ITGB1, and the results indicated that the RSD motif of the F protein interacted with ITGB1 via hydrogen bonds and hydrophobic forces. Further studies showed that ITGB1 plays an important role in aMPV/C infection at the early stage (attachment and internalization). In addition, aMPV/C can infect efficiently human A549 cells, which may be related to sharing the similar structures of F protein-specific motifs (RSD for aMPV/C and RGD for hMPV) recognized by ITGB1. Overall, our results indicate that ITGB1 serves as a functional receptor benefiting from aMPV/C infection by recognizing the RSD motif of the aMPV/C F protein.

Viral attachment to a receptor is a prerequisite for viral infection. We used the CRISPR Cas9 system to knock out ITGB1 in DF-1 cells to investigate the role of ITGB1 in aMPV/C infection. The results found that the ITGB1-knocked-out cells were unable to survive, which is in agreement with that in a previous report [36]. The integrin can combine with ligands for mediating viral attachment and entry such as aMPV/B, Kaposi’s sarcoma-associated herpesvirus, Epstein–Barr virus, and reovirus attachment [37,38,39,40]. Integrin-mediated viral attachment and entry are not only associated with the RGD motif but also bind to the RDD and RSD motifs [33]. Fortunately, a highly conserved RSD motif (329RSD331) was found in the F protein of aMPV/C. aMPV/C F protein induces cell–cell fusion and virus infection [41,42,43], prompting us to explore the role of the RSD motif in viral attachment and entry. RSD and RGD motifs not only play an important role in other pathogens’ infection, such as group A Streptococcus, which can promote binds to human integrins alphavbeta3 and alphaIIbbeta3 (Equine herpesvirus 1 entry via endocytosis is facilitated by alphaV integrins and an RSD motif in glycoprotein D [43,44]); however, they are also involved in cell adhesion and viability [45,46]. In addition to the important role of the conserved domain RSD in aMPV infection, the cleavage of the precursor F protein (F0) is also a prerequisite for F protein-mediated membrane fusion, which, in turn, mediates viral entry and infection [42,47]. The previous research showed that residue 294G influences the cleavage of F0 in a low pH-dependent manner, impacting aMPV/C infectivity and F protein-induced cell–cell fusion [42]. Molecular docking, which can well simulate the binding and force between proteins and peptides [48,49], was used to simulate the binding of the viral F protein to ITGB1, indicating that the aMPV/C F can tightly bind to ITGB1 via hydrogen bonds and hydrophobic forces, thereby promoting aMPV/C infection. ITGB1 can also cooperate with other integrins to promote Ebolavirus infection [40]. In addition, peptide drugs have been widely used in diseases affecting the central nervous system and infectious diseases, and the GRGDSP peptide was used as an alternative to vaccines to prevent Rabies virus infection [50,51]. Since there are currently no drugs targeting aMPV/C infection, peptide drugs could become an alternative. However, the safety and efficacy of RSD peptide against aMPV/C infection need to be further monitored before being recommended for utilization.

Viruses often use conserved receptors like integrin to infect different species. Conserved receptors may provide viruses with an evolutionary advantage to explore alternative hosts, thereby leading to host switching and viral speciation [52]. In a previous study using a reverse genetics system to rescue aMPV/C, researchers found that GFP-expressing aMPV/C and GFP-expressing hMPV could be recovered using the support plasmids of either virus, denoting that the genome promoters are conserved between the two metapneumoviruses and can be cross-recognized by the polymerase complex proteins of either virus [53]. In addition, some research demonstrated that integrin αVβ1, which was composed of integrin αV and ITGB1, is a functional receptor that can promote hMPV infection [32]. Avians have been shown to act as intermediate hosts of some zoonotic viruses [54,55]. Considering the presumed avian origin of metapneumoviruses, the viral functional engagement of conserved receptors may explain the mechanism that enables a virus that is ancestral to metapneumoviruses to break the barrier between birds and humans. However, ITGB1, as a receptor for a potential cross-species transmission of aMPV/C infection, requires further study.

In summary, this study showed that ITGB1 is essential for efficient aMPV/C infection and that the blockade of cell surface ITGB1 inhibits aMPV/C infectivity. The aMPV/C F protein binds to target cells, and the mutation of the RSD motif abolishes this binding. These findings demonstrate the role of ITGB as a receptor for aMPV/C infection and provide a potential therapeutic target and specific small molecule drugs for aMPV/C prevention and treatment.

## 4. Materials and Methods

### 4.1. Cells, Viruses, Antibodies, and Peptides

DF-1 and A549 cells were obtained from the American Type Culture Collection, maintained in our laboratory, and cultured in Dulbecco’s modified Eagle’s medium (DMEM; Gibco, Grand Island, NY, USA) with 10% fetal bovine serum (FBS) (Gibco, NY, USA) supplemented with 1% penicillin and streptomycin. CRFK cells were gifted by Dr. Yuanhong Wang of the Shanghai Veterinary Research Institute, Chinese Academy of Agricultural Sciences. Cells were cultured in Dulbecco’s modified Eagle’s medium (DMEM) (Gibco, NY, USA) with 10% fetal bovine serum (FBS) (Gibco, NY, USA) supplemented with 1% penicillin and streptomycin. IPEC-DQ (DQ) cells, a subclone of the porcine intestinal epithelial cell line IPEC-J2, were a gift from Dr. Dongwan Yoo of the College of Veterinary Medicine, University of Illinois, Urbana–Champaign, Urbana, Illinois, USA. The cells were cultured in RPMI 1640 medium (Gibco, NY, USA) containing 10% FBS (Gibco, NY, USA) supplemented with 1% penicillin and 1% streptomycin at 37 °C in a 5% CO_2_ incubator.

aMPV/C strain JC was isolated from Chinese native meat-type chickens with acute respiratory diseases and maintained in our laboratory (14). All viruses were stored at −80 °C prior to use.

The RSD and control peptides were synthesized using Nanjing Genscript and stored at −20 °C until use. The commercial antibodies were used as follows: integrin αV (MAB2021) and β1 (MAB1959) were purchased from Millipore (Millipore, New York, NY, USA); integrin α6 (100497-T08) and β3 (101954-T32) were purchased from Sino biological (Sino biological, Beijing, China); integrin α2 (MAB228Hu22) was purchased from USCN Life Science (USCN Life Science, Wuhan, China); and horseradish peroxidase (HRP)-conjugated anti-rabbit secondary antibody (AS014), HRP-conjugated anti-mouse secondary antibody (AS003), and mouse anti-GAPDH antibody (AC001) were purchased from ABclone (ABclone, Wuhan, China). The Alexa Fluor 488-conjugated goat anti-mouse antibody (A11001) was purchased from Sigma-Aldrich (Sigma Aldrich, New York, NY, USA). Mouse antiviral N-polyclonal antibody was prepared and stored at our laboratory.

### 4.2. Cell Viability

Cell Counting Kit-8 (Sigma Aldrich, Milwaukee, WI, USA) was used to determine cell viability. DF-1 cells in 96-well plates were incubated with 100 μL of Cell Counting reagent (Sigma Aldrich, WI, USA) per well for 2 h on a shaker to mediate cell lysis. Luminescence was examined using Tecan Infinite M Plex (Tecan, Hombrechtikon, Switzerland).

### 4.3. Transfection of Chicken and Human Integrin cDNAs

Full-length cITGB1 and hITGB1 were isolated from DF-1 and A549 cells, respectively, and cloned into the pCMV-Flag-N vector. All constructed plasmids were confirmed using DNA sequencing, and all primers used for plasmid construction are displayed below: Chicken-ITGB1 (sense, CGCGAATTCGGatggccgagactaatttaacattgctca; antisense, CGCGGTACCtcattttccctcatatttaggattgacca), Human-ITGB1 (sense, CCGGAATTCGGatgaatttacaaccaattttctggattgg; antisense, CGGGGTACCtcattttccctcatacttcggattgacca). Cells grown in six-well plates (Nest, Wuxi, China) were transfected with either control plasmids or plasmids encoding chicken and human integrin constructs using Lipofectamine™ 2000 reagent (11668030, Invitrogen, New York, NY, USA). The cells were incubated for 24 h to allow integrin expression before adsorption with aMPV/C for infectivity studies.

### 4.4. Knockdown of Integrin Expression with siRNA

siRNA targeting chicken integrin β1 was synthesized using GenePharma (GenePharma, Suzhou, China), and the sequences are listed as follows: SiITGB1, sense, GCGAUCGAUCAAACGGUUUTT, antisense, AAACCGUUUGAUCGAUCGCTT. DF-1 cells grown in six-well plates to reach 90% confluence were transfected with control or integrin-specific siRNAs using the Lipofectamine RNAi MAX reagent (13778150, Invitrogen, NY, USA). After incubation for 24 h, the cells were collected for the determination of integrin expression using Western blotting.

### 4.5. Quantitative Real-Time Reverse Transcription PCR (RT-qPCR)

Total RNA was extracted from cells using TRIzol reagent (Pufei, Shanghai, China), and the extracted RNA was reverse-transcribed using the TransScript cDNA Synthesis Kit (TransScript, Nanjing, China) to generate cDNAs. The generated cDNA was analyzed via RT-qPCR using a SuperReal Premix Color SYBR Green I mix (Tiangen, Beijing, China) and quantified using a Light Cycler 96 system (Roche, Basel, Switzerland). The quantitative RT-qPCR method was developed in our laboratory, and all samples were analyzed in triplicate. All RT-qPCR primers utilized in this study are as follows: aMPV/C-M-F, TTGATGAATTGCTGAGAAT; aMPV/C-M-R, AACAACCTTAGTGAACCT.

### 4.6. Virus Attachment and Entry Assays

For the attachment assay, DF-1 cells were inoculated with aMPV/C at an MOI for 2 in the presence of the inhibitors at 4 °C for 1 h. After the incubation period, the inoculum was removed, and the cells were washed three times with cold phosphate-buffered saline (PBS) and then harvested in three freeze-thaw cycles and subjected to viral RNA extraction for the quantification of viral load via real-time RT-PCR. For the entry assay, after the binding period, cells treated with inhibitors or transfected with siRNAs were incubated with a medium for 1 h at 37 °C and washed with cold PBS before being harvested and subjected to viral RNA extraction for the quantification of viral load via real-time PCR (RT-qPCR). We established an absolutely quantitative method to detect virus copy numbers. The sense primer used for detection is GTCAATTCAGCCAAGGCAGT, and the anti-sense primer is GGGGCAATCCTAGCTTGAGT. The standard equation used for calculations is y = −3.427x + 34.94.

### 4.7. Virus Titer Determination

Supernatants from aMPV/C-infected cells were collected and stored at −80 °C before analysis. To determine the aMPV/C titer, the supernatants were 10-fold serially diluted and cultured with DF-1 cells in 96-well plates. Virus titers were determined as previously described (13) and expressed as a 50% tissue culture infective dose (TCID_50_) per 0.1 mL.

### 4.8. Western Blotting

Proteins were extracted from whole cells using a radioimmunoprecipitation assay (RIPA) and NP40 lysis buffer (Beyotime, Shanghai, China) containing 1 mM phenylmethanesulfonyl fluoride (PMSF) (Beyotime, Shanghai, China). The extracted proteins were quantified using the BCA kit (Beyotime, Shanghai, China). The cell lysates were quantified, resolved using standard sodium dodecyl sulfate-polyacrylamide gel electrophoresis, and blotted onto nitrocellulose membranes (Pall, New York, NY, USA). The membranes were blocked with 5% non-fat milk and incubated with the corresponding primary and secondary antibodies. Immunoreactive bands were visualized using the AMERSHAM ImageQuant800 chemiluminescence imaging system (GE, Chicago, IL, USA).

### 4.9. Immunostaining Assay of Viral Infection

After being washed with PBS, cells were fixed with 4% paraformaldehyde (PFA), blocked by PBS with 5% nonfat milk, and subsequently incubated with the corresponding primary and secondary antibodies (Alexa Fluor 488-conjugated goat α-mouse antibody), followed by incubation with 4′,6-diamidino-2-phenylindole (DAPI) (Sigma Aldrich, WI, USA). The cells were examined under an inverted fluorescence microscope (Olympus Corporation, Tokyo, Japan).

### 4.10. Artificial Intelligence Predicts Protein Structure and Molecular Docking

The structures of the aMPV/C F protein and chicken and human ITGB1 were predicted using the artificial intelligence algorithm AlphaFold2 deployed on a supercomputer (BKUNYUN CLOUD, Shenzhen, China). For molecular docking, the RSD peptide structure was first mapped using ChemBioDraw Ultra 14.0, and the mapped peptide structures were imported into ChemBio3D Ultra 14.0 for energy minimization. The RMS gradient limit was established as 0.001, and the small molecules were saved in mol2 format. The optimized small molecules were entered using AutodockTools (v1.5.6) for hydrogenation, charge distribution, charge calculation, and rotatable bond setting, and saved in pdbqt format. The prepared ITGB1 protein was imported into Pymol2.3.0 to delete the original ligand and protein crystal water, and the protein structure was entered using AutoDocktools for hydrogenation, charge assignment, charge calculation, atom type specification, and saving in pdbqt format. POCASA 1.1 was utilized to predict binding sites, and AutoDock Vina1.1.2 was utilized for docking. Center-x = 35.1, center-y = 58.0, center-z = −52.5. Search space: size-x: 80, size-y: 80, size-z: 80 (the spacing of each grid point is 0.375 Å), exhaustiveness: 10. All other parameters were set by default. PyMOL 2.3.0 and LIGPLOT (v2.2.4) were utilized to analyze the interaction manner of the docking results.

### 4.11. Phylogenetic Analysis

A maximum-likelihood tree was constructed based on the different host species of the F gene using IQtree software (V2.2.2.6) with the GTRGAMMAI model and 1000 replicates.

### 4.12. Statistical Analysis

Data are expressed as means ± standard deviation (SD). Differences were determined utilizing a one-way analysis of variance or Student’s *t*-test, utilizing the GraphPad Prism 9.0 software (GraphPad Software, La Jolla, CA, USA). *p* value < 0.05 was considered statistically significant.

## Figures and Tables

**Figure 1 ijms-25-00829-f001:**
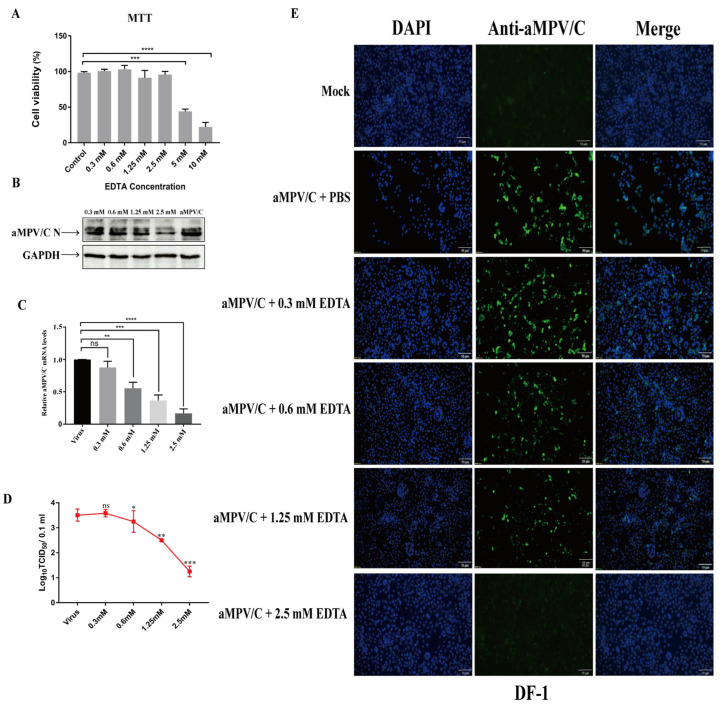
EDTA effectively inhibits infection through aMPV/C in cultured DF-1 cells: (**A**) Cell viability was analyzed after treatment with different concentrations of EDTA (0.3, 0.6, 1.25, 2.5, 5, and 10 mM) using an MTT assay. (**B**) DF-1 cells were treated with different concentrations of EDTA (0.3, 0.6, 1.25, and 2.5 mM) at 4 °C for 1 h before viral adsorption; then, infected cells were incubated in a medium supplemented with 2% FBS for 72 h. Western blotting was used to quantify aMPV/C N protein expression, and GAPDH was used as a control. (**C**) RT-qPCR was used to determine the relative expression level of aMPV/C RNA in DF-1 cells after EDTA treatment. Expression was normalized to GAPDH mRNA. (**D**) Viral titers in the supernatants of aMPV/C-infected cells with different concentrations of EDTA treatment were determined 72 h post-infection using a TCID_50_ assay. (**E**) DF-1 cells were infected with aMPV/C after EDTA treatment. The cells were incubated with antibodies corresponding to aMPV/C N protein followed by fluorescein isothiocyanate (FITC)-conjugated anti-mouse IgG antibody (green), and nuclei were incubated with DAPI (blue). The cells were examined under an inverted fluorescence microscope. Data are expressed as means ± SD from triplicate independent experiments (ns, not significant; *, *p* < 0.05, **, *p* < 0.01; ***, *p* < 0.001; ****, *p* < 0.0001).

**Figure 2 ijms-25-00829-f002:**
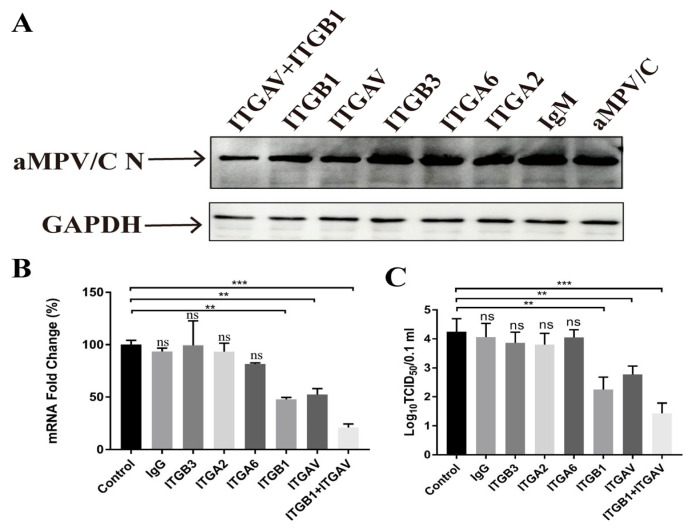
Integrin function-blocking antibodies inhibit aMPV/C infection: (**A**) DF-1 cells were incubated with blocking antibodies to integrin or control IgG treatment before adsorption with aMPV/C and incubation in a medium supplemented with 2% FBS for 72 h. All antibodies were used at 20 μg/mL. Western blotting was utilized to quantify the expression level of the viral protein, and GAPDH was used as a control. (**B**) RT-qPCR was used to determine the aMPV/C M gene RNA level in DF-1 cells after antibody inhibition treatment. Expression was normalized to GAPDH mRNA. (**C**) Viral productions in the supernatants of aMPV/C-infected cells with different concentrations of EDTA treatment were determined 72 h post-infection using a TCID_50_ assay. Data are shown as means ± SD from triplicate independent experiments (**, *p* < 0.01; ***, *p* < 0.001, ns, Not significant).

**Figure 3 ijms-25-00829-f003:**
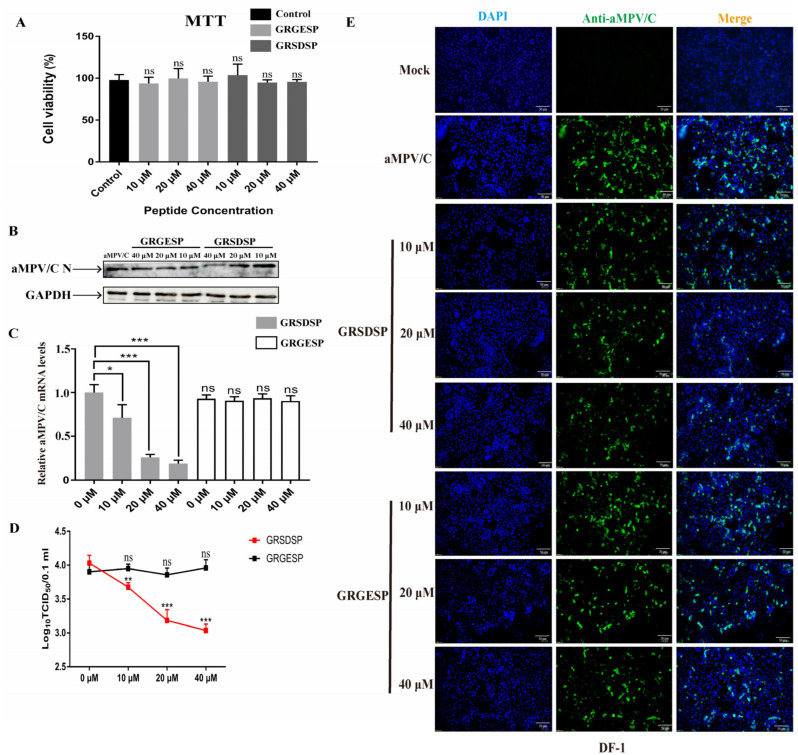
RSD peptide effectively inhibits infection through aMPV/C in cultured DF-1 cells: (**A**) Cell viability was analyzed after treatment with different concentrations of the peptides GRGESP or GRSDSP using an MTT assay. (**B**) Different concentrations of GRSDSP (10, 20, and 40 μM) or the control peptide GRGESP (10, 20, and 40 μM) were mixed with aMPV/C and incubated at 4 °C for 1 h; then, infected cells were incubated in a medium supplemented with 2% FBS for 72 h. Western blotting was used to quantify the protein expression level of aMPV/C N after GRSDSP and GRGESP peptide treatment, and GAPDH was used as a control. (**C**) RT-qPCR was used to analyze the aMPV/C RNA level in DF-1 cells after peptide treatment. Expression was normalized to GAPDH mRNA. (**D**) Viral titers in the supernatants of aMPV/C-infected cells under different peptide treatments were determined 72 h post-infection using a TCID_50_ assay. (**E**) Cells were infected with aMPV/C after GRSDSP and GRGESP peptide treatment and incubated in a medium supplemented with 2% FBS for 72 h. The cells were incubated with antibodies against aMPV/C N protein, followed by incubation with FITC-conjugated goat anti-mouse IgG (green) secondary antibodies. Nuclei were incubated with DAPI (blue), and the cells were observed under an inverted fluorescence microscope. Data are expressed as means ± SD from triplicate independent experiments (ns, not significant; *, *p* < 0.05; **, *p* < 0.01; ***, *p* < 0.001).

**Figure 4 ijms-25-00829-f004:**
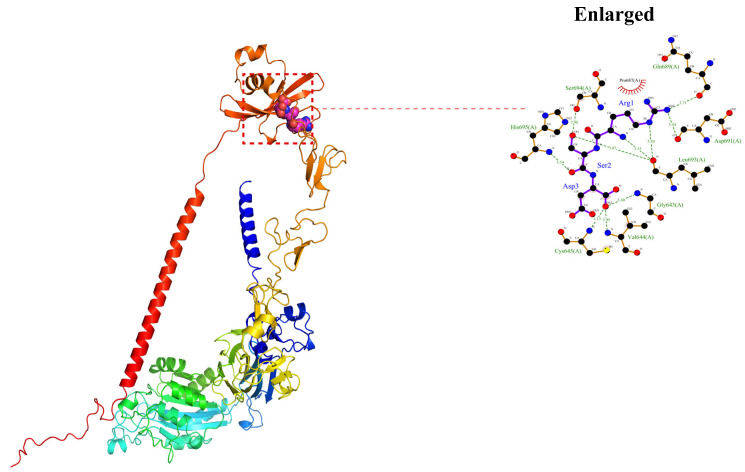
aMPV/C F protein binds cITGB1. POCASA 1.1 was utilized to predict binding sites, and AutoDock Vina1.1.2 was utilized for docking. aMPV/C F protein could bind ITGB1, and its main binding region was located in the 643–695 region of ITGB1.

**Figure 5 ijms-25-00829-f005:**
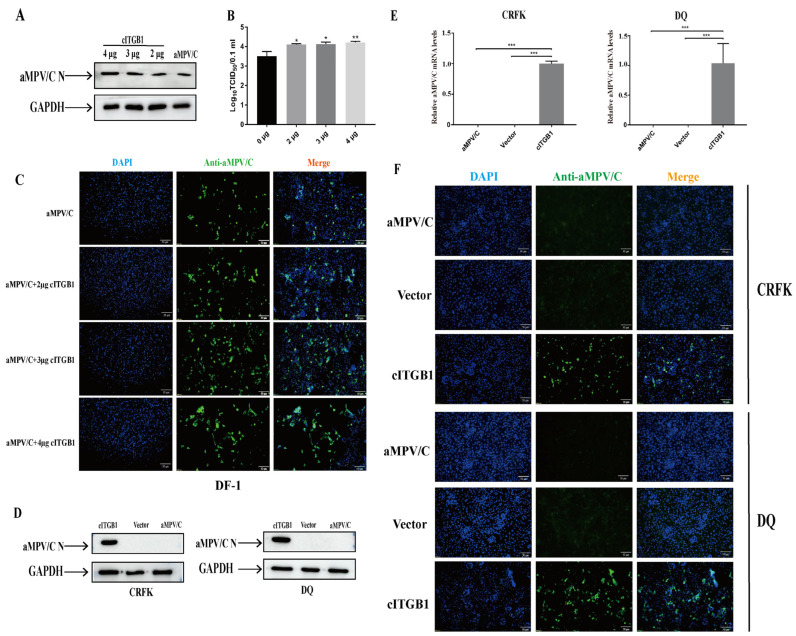
Overexpression of cITGB1 promotes aMPV/C infection in DF-1 cells: (**A**,**B**) Cells were transfected with recombinant plasmid cITGB1 for 24 h and subsequently infected with aMPV/C for 72 h. The aMPV/C replication levels were detected using Western blotting (**A**) and a TCID_50_ assay (**B**). (**C**) Cells were infected with aMPV/C after transfection with cITGB1. The cells were incubated with antibodies against aMPV/C N protein, followed by incubation with FITC-conjugated goat anti-mouse IgG (green) secondary antibodies. Nuclei were incubated with DAPI (blue), and the cells were observed under an inverted fluorescence microscope. (**D**,**E**) CRFK and DQ cells were transfected with recombinant plasmid cITGB1 for 24 h, incubated with the integrin-specific antibody, and, subsequently, infected with aMPV/C for 72 h. aMPV/C replication levels were detected using Western blotting (**D**) and RT-qPCR (**E**). (**F**) DQ and CRFK cells were infected with aMPV/C after transfection with cITGB1. The cells were incubated with antibodies against aMPV/C N protein, followed by incubation with FITC-conjugated goat anti-mouse IgG (green) secondary antibodies. Nuclei were incubated with DAPI (blue), and the cells were examined under an inverted fluorescence microscope. Data are shown as means ± SD from triplicate independent experiments (*, *p* < 0.05; **, *p* < 0.01; ***, *p* < 0.001).

**Figure 6 ijms-25-00829-f006:**
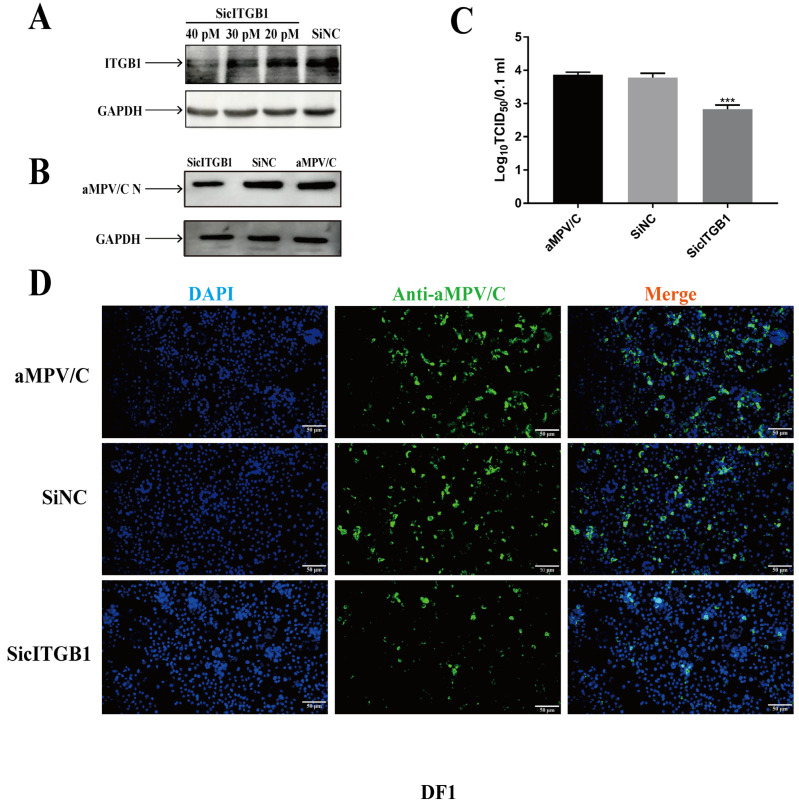
Knockdown of cITGB1 expression inhibits aMPV/C infection: DF-1 cells were transfected with siRNA targeting cITGB1 or negative control siRNA for 24 h and subsequently infected with aMPV/C for 72 h. (**A**) Knockdown effects of different concentrations of siRNA on ITGB1. (**B**) Western blotting was used to analyze the viral protein expression level after siRNA knockdown, and GAPDH was used as a control. (**C**) Viral titers in the supernatants of aMPV/C-infected cells after siRNA knockdown were determined 72 h post-infection using TCID_50_ assay. (**D**) DF-1 cells were transfected with siRNA targeting cITGB1 or negative control siRNA and subsequently infected with aMPV/C for 72 h. The cells were incubated with antibodies against aMPV/C N protein, followed by incubation with FITC-conjugated goat anti-mouse IgG (green) secondary antibodies. Nuclei were incubated with DAPI (blue), and the cells were examined under an inverted fluorescence microscope. Data are shown as means ± SD from triplicate independent experiments (***, *p* < 0.001).

**Figure 7 ijms-25-00829-f007:**
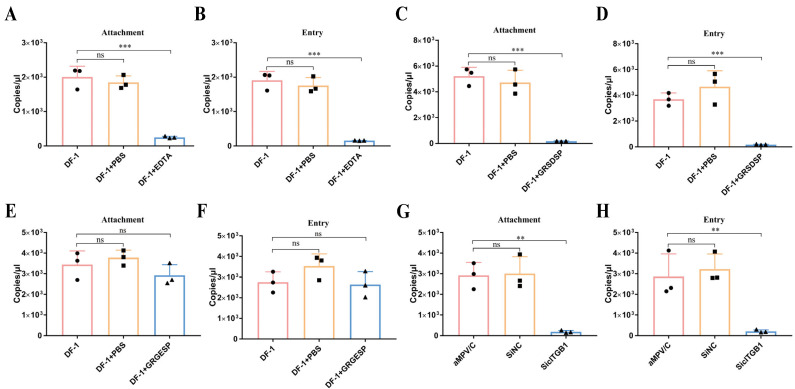
aMPV/C relies on ITGB1 adsorption on the cell surface: (**A**,**B**) EDTA inhibited aMPV/C attachment. DF-1 cells were treated using EDTA (2.5 mM) at 4 °C for 1 h before viral adsorption. After that, the EDTA-treated DF-1 cells were infected by aMPV/C strain JC for 1 h at 4 °C (virus attachment), followed by the removal of unbound viruses, then incubated at 37 °C for 1 h (virus entry), and viral load was detected using RT-qPCR. (**C**–**F**) RSD peptide (GRSDSP) blocked aMPV/C attachment in DF-1 cells. GRSDSP peptide or control peptide (40 μM) was mixed with aMPV/C at an MOI of 2 at 4 °C for 1 h. DF-1 cells were infected with a mixture of peptide and aMPV/C for 1 h at 4 °C (virus attachment) (**C**,**E**), followed by removal of unbound viruses, then incubated at 37 °C for 1 h (virus entry) (**D**,**F**). (**G**,**H**) Knockdown of the ITGB1 expression can inhibit aMPV/C attachment. DF-1 cells were transfected with siRNA-targeting cITGB1 or negative control siRNA for 24 h; after that, the cells were infected with aMPV/C for 1 h at 4 °C for detecting the virus attachment level (**G**), followed by the removal of unbound viruses, then incubated at 37 °C for 1 h for detecting the virus entry level (**H**) using RT-qPCR (**, *p* < 0.01; ***, *p* < 0.001, ns, not significant).

**Figure 8 ijms-25-00829-f008:**
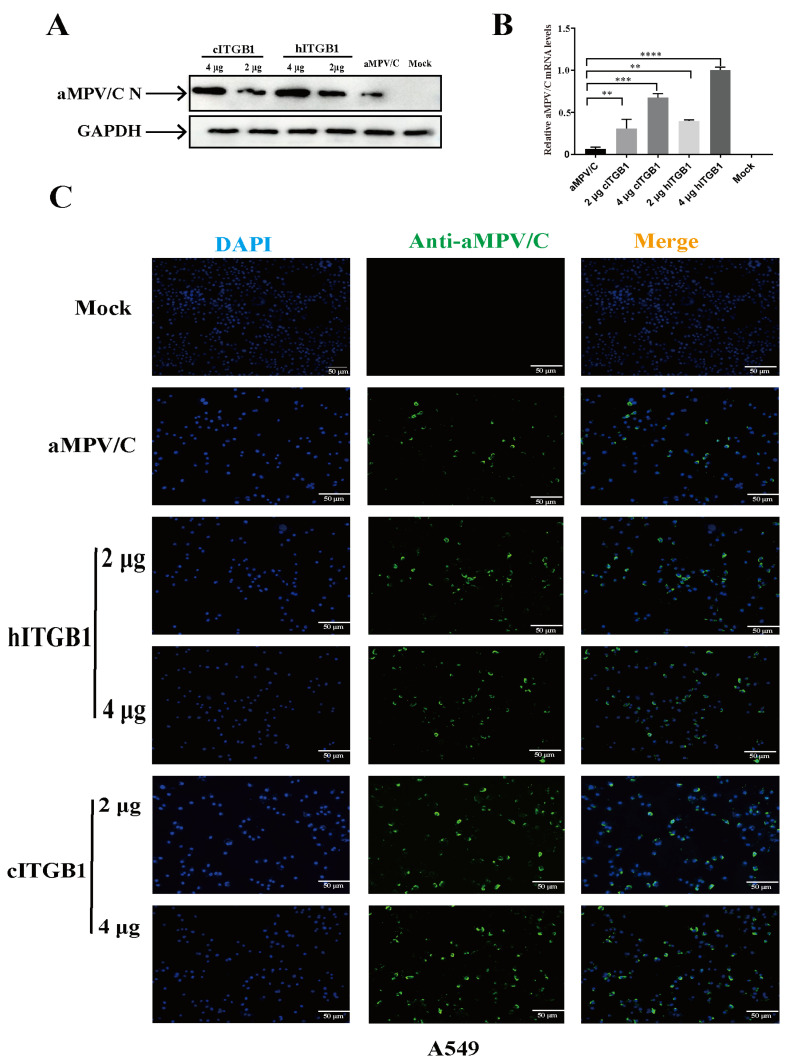
ITGB1 promotes aMPV/C infection in human cells: A549 cells were transfected with recombinant plasmid cITGB1 or hITGB1 for 24 h and subsequently infected with aMPV/C for 72 h. (**A**) Western blotting was used to analyze the viral protein expression level after transfection with cITGB1 or hITGB1 in A549 cells, and GAPDH was used as a control. (**B**) RT-qPCR was used to analyze the relative aMPV/C RNA levels in A549 cells after transfection with cITGB1 or hITGB1. Expression was normalized to GAPDH mRNA. (**C**) After transfection with the expression plasmid cITGB1 or hITGB1, A549 cells were infected with aMPV/C for 72 h. Subsequently, the cells were incubated with antibodies against aMPV/C N protein, followed by incubation with FITC-conjugated goat anti-mouse IgG (green) secondary antibodies. Nuclei were incubated with DAPI (blue), and the cells were examined under an inverted fluorescence microscope. (**, *p* < 0.01; ***, *p* < 0.001, ****, *p* < 0.0001).

## Data Availability

Data is contained within the article and Appendix A.

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
