# Peer review of "A Receptor Integrin β1 Promotes Infection of Avian Metapneumovirus Subgroup C by Recognizing a Viral Fusion Protein RSD Motif"

_ijms, 2024, doi:10.3390/ijms25020829_

Round 1
Reviewer 1 Report
Comments and Suggestions for Authors
This was an interesting paper about identifying integrin B1 as required for aMPV/C infection. I believe this paper should be accepted for publication pending addressing a few minor things:
1. I would like a fuller understanding/explanation of what the role of the F protein vs the G protein are for entry and fusion.
What is the function of the G protein in this vs the F protein?
This would be helpful for a broader audience because most highly studied virus people are familiar with only utilized 1 protein for this cellular entry and fusion – Such as glycoprotein (G) for VSV, HA for influenza, and Spike (S) for SARS-CoV-2,
2. Images such as Figure 1E, 3E, 5C/F, 6D, 8C, are hard to see. If would be helpful if the contrast was adjusted to make these pictures brighter. Also, cropping the pictures to zoom in on a smaller region would also be helpful where you can better make out the cells.
3. Proteolytic processing and pH have been shown to be important for the F protein of human Metapneumovirus (Schowalter et al. 2006 JVI) and aMPV/C (Yun et al. 2015 Sci Reports). This suggest that entry of this virus through an endosomal compartment. While the integrin binding is likely important for cellular attachment and endocytosis. Could this be further clarified and explained. Also could the role of pH be verified with in the context of your experiments by maybe testing BafA1 on effecting infectivity.
Author Response
Dear editor and reviewers,
Thank you very much for the chance you gave us to revise our manuscript. We thank the reviewers for the time and effort that they have put into reviewing the previous version of the manuscript. We admire your expertise and patience. The comments are very useful for our research and publication. We decide to accept all the comments and revise our manuscript carefully according to your comments. The changes were listed blew point by point, and were marked in red in the revised manuscript. The manuscript has greatly benefited from these invaluable suggestions.
Based on the instructions provided in your letter, we uploaded the file of the revised manuscript. Appended to this letter is our point-by-point response to the comments raised by the reviewers.
We look forward to working with you to move this manuscript closer to publication in International Journal of Molecular Science.
Reviewer 1:
This was an interesting paper about identifying integrin B1 as required for aMPV/C infection. I believe this paper should be accepted for publication pending addressing a few minor things:
- I would like a fuller understanding/explanation of what the role of the F protein vs the G protein are for entry and fusion. What is the function of the G protein in this vs the F protein? This would be helpful for a broader audience because most highly studied virus people are familiar with only utilized 1 protein for this cellular entry and fusion – Such as glycoprotein (G) for VSV, HA for influenza, and Spike (S) for SARS-CoV-2,
Response: Thanks for your good suggestion. Different from most viruses such as glycoprotein (G) for VSV, HA for influenza, and Spike (S) for SARS-CoV-2, avian metapneumovirus uses three viral proteins to help its cellular entry and fusion. The G protein mainly plays the role of mediating the adsorption of aMPV to host cells, SH plays an important role in cell-cell fusion, while the F protein can bind with receptor which mediate the fusion of the viral envelope and the cell membrane without relying on G and SH proteins. The relevant descriptions and literatures have been added to Page 2 lines 56-64 and Page 17 lines 767-784 of the revised manuscript, respectively.
- Images such as Figure 1E, 3E, 5C/F, 6D, 8C, are hard to see. If would be helpful if the contrast was adjusted to make these pictures brighter. Also, cropping the pictures to zoom in on a smaller region would also be helpful where you can better make out the cells.
Response: Thanks for your suggestion. All images, including Figure 1E, 3E, 5C/F, 6D, and 8C, have been adjusted, you can see them on figures of the revised manuscript.
- Proteolytic processing and pH have been shown to be important for the F protein of human Metapneumovirus (Schowalter et al. 2006 JVI) and aMPV/C (Yun et al. 2015 Sci Reports). This suggest that entry of this virus through an endosomal compartment. While the integrin binding is likely important for cellular attachment and endocytosis. Could this be further clarified and explained. Also could the role of pH be verified with in the context of your experiments by maybe testing BafA1 on effecting infectivity.
Response: Thanks for your good suggestions. Cleavage of the F protein is critical for metapneumovirus infection. Previous research data has shown that residue 294G influences the cleavage of F0 in a low pH-dependent manner, further impacting on aMPV infectivity and F protein-induced cell-cell fusion. The relevant descriptions and literatures have been inserted into Page 12 lines 550-555, Page 18 lines 819-821, and Page 19 lines 831-832 of the revised manuscript, respectively.

Reviewer 2 Report
Comments and Suggestions for Authors
In this manuscript, the authors demonstrated that ITGB1 acts as an essential receptor for aMPV/C attachment and internalization into host cells in which facilitates aMPV/C infection via different treatment, such as siRNA, peptides, chemical reagent, and non-permissive cells. The experimental design and data are of integrity and well presented in this manuscript. However, there are some concerns to be addressed as listed in the following comments.
1. Please add rulers on all fluorescent images.
2. The statement of “mRNA fold change (%)” needs to be standardly described in all Figures.
3. In Fig. 1, 3, 5, 6, 8, is aMPV/C in green color stands for anti-aMPV/C N protein? If yes, please label clearly for the reader because it is confused that aMPV/C in the Left side and Top side.
4. In Fig.1, “*, P<0.01” could be missed for Fig 1D showed a * at 0.6mM.
5. In Line 296-298, “The enhanced efficiency of aMPV/C infection in chicken β1 integrin-transfected DQ and CRFK cells was inhibited by preincubation with blocking antibodies for β1 integrin (Fig. 5D-5F)”. Can’t find the inhibition results from Fig. 5D-5F, would the authors indicate it more clearly.
6. In Fig. 5D, the labeling of different groups may be incorrect in DQ cells.
7. In Fig. 7, the primers of absolute quantitative primers were not listed in table, and the related method is not described.
8. There are some words that a space needed. For example, Line 515. Please check the whole draft once again.
Author Response
Dear editor and reviewers,
Thank you very much for the chance you gave us to revise our manuscript. We thank the reviewers for the time and effort that they have put into reviewing the previous version of the manuscript. We admire your expertise and patience. The comments are very useful for our research and publication. We decide to accept all the comments and revise our manuscript carefully according to your comments. The changes were listed blew point by point, and were marked in red in the revised manuscript. The manuscript has greatly benefited from these invaluable suggestions.
Based on the instructions provided in your letter, we uploaded the file of the revised manuscript. Appended to this letter is our point-by-point response to the comments raised by the reviewers.
We look forward to working with you to move this manuscript closer to publication in International Journal of Molecular Science.
Reviewer 2:
In this manuscript, the authors demonstrated that ITGB1 acts as an essential receptor for aMPV/C attachment and internalization into host cells in which facilitates aMPV/C infection via different treatment, such as siRNA, peptides, chemical reagent, and non-permissive cells. The experimental design and data are of integrity and well presented in this manuscript. However, there are some concerns to be addressed as listed in the following comments.
- Please add rulers on all fluorescent images.
Response: Thanks for your good suggestions. Scale bars have been added to all figures in the revised manuscript.
- The statement of “mRNA fold change (%)” needs to be standardly described in all Figures.
Response: Thank you for your good suggestions. The statement of “mRNA fold change (%)” in all figures have been modified in the revised manuscript.
- In Fig. 1, 3, 5, 6, 8, is aMPV/C in green color stands for anti-aMPV/C N protein? If yes, please label clearly for the reader because it is confused that aMPV/C in the Left side and Top side.
Response: Thanks for your good suggestions. aMPV/C in green color in Figures 1, 3, 5, 6, and 8 stands for anti-aMPV/C N protein, and modifications have been made in the revised manuscript.
- In Fig.1, “*, P<01” could be missed for Fig 1D showed a * at 0.6mM.
Response: Thanks for your good suggestions. The missing information has been added to Page 4 line 156 of the revised manuscript.
- In Line 296-298, “The enhanced efficiency of aMPV/C infection in chicken β1 integrin-transfected DQ and CRFK cells was inhibited by preincubation with blocking antibodies for β1 integrin (Fig. 5D-5F)”. Can’t find the inhibition results from Fig. 5D-5F, would the authors indicate it more clearly.
Response: Thanks for your good suggestions. The modification has been made on Page 7 lines 296-299 of the revised manuscript.
- In Fig. 5D, the labeling of different groups may be incorrect in DQ cells.
Response: The errors in Figure 5 have been corrected in the revised manuscript.
- In Fig. 7, the primers of absolute quantitative primers were not listed in table, and the related method is not described.
Response: The missing primers and related method have been inserted into Page 14 lines 652-656 of the revised manuscript.
- There are some words that a space needed. For example, Line 515. Please check the whole draft once again.
Response: We have rechecked and corrected in the revised manuscript.
